

# Evidence for widespread dysregulation of circadian clock progression in human cancer

Jarrod Shilts[1], Guanhua Chen[2] and Jacob J. Hughey[1,3]

[1] Department of Biomedical Informatics, Vanderbilt University School of Medicine, Nashville, TN, United States of America
[2] Department of Biostatistics & Medical Informatics, University of Wisconsin-Madison, Madison, WI, United States of America
[3] Department of Biological Sciences, Vanderbilt University, Nashville, TN, United States of America

## ABSTRACT

The ubiquitous daily rhythms in mammalian physiology are guided by progression of the circadian clock. In mice, systemic disruption of the clock can promote tumor growth. *In vitro*, multiple oncogenes can disrupt the clock. However, due to the difficulties of studying circadian rhythms in solid tissues in humans, whether the clock is disrupted within human tumors has remained unknown. We sought to determine the state of the circadian clock in human cancer using publicly available transcriptome data. We developed a method, called the clock correlation distance (CCD), to infer circadian clock progression in a group of samples based on the co-expression of 12 clock genes. Our method can be applied to modestly sized datasets in which samples are not labeled with time of day and coverage of the circadian cycle is incomplete. We used the method to define a signature of clock gene co-expression in healthy mouse organs, then validated the signature in healthy human tissues. By then comparing human tumor and non-tumor samples from twenty datasets of a range of cancer types, we discovered that clock gene co-expression in tumors is consistently perturbed. Subsequent analysis of data from clock gene knockouts in mice suggested that perturbed clock gene co-expression in human cancer is not caused solely by the inactivation of clock genes. Furthermore, focusing on lung cancer, we found that human lung tumors showed systematic changes in expression in a large set of genes previously inferred to be rhythmic in healthy lung. Our findings suggest that clock progression is dysregulated in many solid human cancers and that this dysregulation could have broad effects on circadian physiology within tumors. In addition, our approach opens the door to using publicly available data to infer circadian clock progression in a multitude of human phenotypes.

Corresponding author
Jacob J. Hughey,
jakejhughey@gmail.com

## INTRODUCTION

Daily rhythms in mammalian physiology are guided by a system of oscillators called the circadian clock (*Dibner, Schibler & Albrecht, 2010*). The core clock consists of feedback loops between several genes and proteins, and based on work in mice, is active in nearly

every tissue in the body (*Yoo et al., 2004*; *Zhang et al., 2014*). The clock aligns itself to environmental cues, particularly cycles of light-dark and food intake (*Damiola et al., 2000*; *Asher et al., 2010*; *Eckel-Mahan et al., 2013*). In turn, the clock regulates various aspects of metabolism (*Nakahata et al., 2009*; *Cho et al., 2012*; *Neufeld-Cohen et al., 2016*; *Guerrero-Vargas et al., 2017*) and is tightly linked to the cell cycle (*Matsuo et al., 2003*; *Gréchez-Cassiau et al., 2008*; *Geyfman et al., 2012*; *Feillet et al., 2014*; *Bieler et al., 2014*; *Matsu-Ura et al., 2016*; *El-Athman et al., 2017*).

Consistent with the tight connections between the circadian clock, metabolism, and the cell cycle, multiple studies have found that systemic disruption of the circadian system can promote cancer. In humans, long-term rotating shift work and night shift work, which perturb sleep-wake and circadian rhythms, have been associated with breast, colon, and lung cancer (*Schernhammer et al., 2001*; *Schernhammer et al., 2003*; *Schernhammer et al., 2013*; *Wegrzyn et al., 2017*). In mice, environmental disruption of the circadian system (e.g., through severe and chronic jet lag) increases the risk of breast cancer and hepatocellular carcinoma (*Van Dycke et al., 2015*; *Kettner et al., 2016*). Furthermore, both environmental and genetic disruption of the circadian system promote tumor growth and decrease survival in a mouse model of human lung adenocarcinoma (*Papagiannakopoulos et al., 2016*). While these studies support the link from the clock to cancer, other studies have established a link in the other direction, whereby multiple components of a tumor, including the RAS and MYC oncogenes, can induce dysregulation of the circadian clock (*Relógio et al., 2014*; *Michael et al., 2015*; *Altman et al., 2015*). Despite this progress, however, whether the clock is actually disrupted within human tumors has remained unclear. Given recent findings that stimulating the circadian clock slows tumor growth in a mouse model of melanoma (*Kiessling et al., 2017*), it is important to determine the extent of clock disruption across human cancers, in order to delineate the general potential of anti-tumor strategies based on restoring or improving clock function.

When the mammalian circadian clock is progressing or "ticking" normally, clock genes and clock-controlled genes show characteristic rhythms in expression throughout the body and *in vitro* (*Balsalobre, Damiola & Schibler, 1998*; *Yoo et al., 2004*; *Zhang et al., 2014*). These rhythms can be used to monitor clock progression in humans (*Laing et al., 2017*; *Hughey, 2017*). Measurements of circadian rhythms through time-course experiments have revealed that the clock is altered or perturbed in some human breast cancer cell lines (*Rossetti et al., 2012*; *Xiang et al., 2012*). Computational methods for analyzing rhythmicity require that samples be labeled with time of day (or time since start of experiment) and acquired throughout the 24-h cycle (*Hughey, Hastie & Butte, 2016*; *Thaben & Westermark, 2016*; *Wu et al., 2016*). Unfortunately, existing data from resected human tumors meet neither of these criteria. In this scenario, one approach might be to look for associations between the expression levels of clock genes and other biological and clinical variables. For example, in human breast cancer, the expression levels of several clock genes have been associated with metastasis-free survival (with the direction of association depending on the gene) (*Cadenas et al., 2014*). However, because a functional circadian clock is marked less by the levels of gene expression than by rhythms in gene expression, this type of analysis cannot necessarily be used to determine whether the clock is progressing normally.

A more sophisticated approach is to assume the presence of rhythms and to infer a cyclical ordering of samples, using methods such as Oscope or CYCLOPS (*Leng et al., 2015*; *Anafi et al., 2017*). By applying CYCLOPS to transcriptome data from hepatocellular carcinoma, *Anafi et al. (2017)* found evidence for weaker or disrupted rhythmicity of several clock genes, as well as genes involved in apoptosis and JAK-STAT signaling, in tumor samples compared to non-tumor samples. Although CYCLOPS does not require that samples be labeled with time of day, it does require that the samples cover the entire cycle. Consequently, the authors recommend that CYCLOPS be applied to datasets from humans with at least 250 samples (*Anafi et al., 2017*). In addition, although CYCLOPS can be used to infer rhythmicity in the expression of individual genes, it is not designed to quantify differences in the overall pattern of those rhythms (e.g., the relative phasing between genes) between conditions (e.g., healthy and diseased).

Rather than attempting to infer an oscillation, an alternative approach might be to take advantage of the pattern of co-expression (e.g., pairwise correlation) that results from different clock genes having rhythms with different phases. Indeed, a previous study found different levels of co-expression between a few clock genes in different subtypes and grades of human breast cancer (*Cadenas et al., 2014*). Although this finding was an important first step, its generalizability has been limited because the correlations in expression were not examined for all clock genes, in other human cancer types, or in healthy tissues where the circadian clock is known to be functional. Thus, a definitive answer to whether the circadian clock is progressing normally in human tumors is still lacking.

In this study, we developed a computational method to characterize the extent of dysregulation of circadian clock progression in human cancer. Using transcriptome data from mice, we defined a robust signature of clock progression based on the co-expression of clock genes. We validated the signature using transcriptome data from various organs in humans, then examined the extent to which the signature was perturbed in tumor compared to non-tumor samples from The Cancer Genome Atlas (TCGA) and from multiple independent datasets. Our findings suggest that dysregulation of circadian clock progression is present in a wide range of human cancers, is not caused solely by the inactivation of core clock genes, and is accompanied by systematic changes in broader circadian gene expression.

## MATERIALS AND METHODS

### Study design

The main goals of this study were to develop a co-expression-based signature of the circadian clock, to validate the signature in healthy human organs, and to use the signature to infer the extent of normal circadian clock progression in human cancer. We focused on transcriptome data because of its wide availability. We selected the datasets of circadian gene expression in mice (both for defining the reference signature and for comparing clock gene knockouts to wild-type) to represent multiple organs, light-dark regimens, and microarray platforms.

We selected the datasets from healthy human organs to include as many circadian studies as possible and to include a range of organs. The dataset from human skin consisted

of samples taken at only three time-points for each subject (9:30 am, 2:30 pm, and 7:30 pm). Datasets from human blood consisted of multiple samples taken throughout the 24-h cycle for each subject. Datasets from human brain were based on postmortem tissue from multiple anatomical areas, and zeitgeber time for each sample was calculated using the respective donor's date and time of death and geographic location. Datasets from cells cultured *in vitro* were based on time-courses following synchronization by dexamethasone, serum, or alternating temperature cycles. For GSE45642 (human brain), we only included samples from control subjects (i.e., we excluded subjects with major depressive disorder).

For datasets of human cancer, we analyzed data from the Cancer Genome Atlas (TCGA), as it contains samples from various cancer types. To complement the breadth of cancer types in TCGA data, we further included human liver cancer and lung cancer datasets from NCBI GEO. We analyzed human cancer datasets that had at least 30 non-tumor samples. When analyzing clock gene expression in human cancer, unless otherwise noted, we included all tumor and non-tumor samples, not just those from patients from whom both non-tumor and tumor samples were collected. For details of the datasets, all of which are publicly available, see Table S1.

## Processing the gene expression data

For TCGA samples, we obtained the processed RNA-seq data (in units of transcripts per million, TPM, on a gene-level basis) and the corresponding metadata (cancer type, patient ID, etc.) from GSE62944 (*Rahman et al., 2015*). For E-MTAB-3428, we downloaded the RNA-seq read files from the European Nucleotide Archive, used Salmon to quantify transcript-level abundances in units of TPM (*Patro et al., 2017*), then used the mapping between Ensembl Transcript IDs and Entrez Gene IDs to calculate gene-level abundances.

For the remaining datasets, raw (in the case of Affymetrix) or processed microarray data were obtained from NCBI GEO and processed using MetaPredict, which maps probes to Entrez Gene IDs and performs intra-study normalization and log-transformation (*Hughey & Butte, 2015*). MetaPredict processes raw Affymetrix data using RMA and customCDFs (*Irizarry et al., 2003*; *Dai et al., 2005*). As in our previous study, we used ComBat to reduce batch effects between anatomical areas in human brain and between subjects in human blood (*Johnson, Li & Rabinovic, 2007*; *Hughey & Butte, 2016*). We also used ComBat to normalize expression between between subjects in the circadian human skin dataset (Fig. S8). ComBat is an empirical Bayes method to standardize the mean and variance of gene expression values between groups of samples.

## Analyzing clock gene expression and co-expression

We first focused on the expression of 12 genes that are considered part of the core clock or are directly controlled by the clock and that show strong, consistently phased rhythms in multiple mouse organs (*Zhang et al., 2014*; *Hughey, Hastie & Butte, 2016*). We calculated times of peak expression (Fig. S2) and strengths of circadian rhythmicity of expression in wild-type and knockout mice (Fig. S15) using ZeitZeiger (*Hughey, Hastie & Butte, 2016*), with three knots for the periodic smoothing splines (*Helwig & Ma, 2014*). ZeitZeiger uses the time of day information for each sample.

We quantified the relationship between expression values of pairs of genes using the Spearman correlation (rho), which is rank-based and therefore invariant to monotonic transformations such as the logarithm and less sensitive to outliers than the Pearson correlation. Using the biweight midcorrelation (*Song, Langfelder & Horvath, 2012*), which is also robust to outliers, gave very similar results. All heatmaps of clock gene co-expression have the same mapping of correlation value to color, so they are directly visually comparable.

For a group of samples from a given dataset, we estimated the strength of clock gene co-expression (Fig. S7) as the difference between the 95th and 5th percentiles of the distribution of Spearman correlations between pairs of the 12 clock genes.

We calculated the reference Spearman correlation for each pair of clock genes (Table S2) using a fixed-effects meta-analysis (*Hedges & Olkin, 1985*) of the eight mouse datasets shown in Fig. S1. First, we applied the Fisher z-transformation (*arctanh*) to the correlations from each dataset. Then we calculated a weighted average of the transformed correlations, where the weight for dataset $i$ was $n_i - 3$ (corresponding to the inverse variance of the transformed correlation), where $n_i$ is the number of samples in dataset $i$. Finally, we applied the inverse transformation (*tanh*) to the weighted average.

To quantify the similarity in clock gene co-expression between the reference and a group of samples from another transcriptome dataset, we calculated the Euclidean distance between the respective Spearman correlation vectors, which contain all values in the strictly lower (or strictly upper) triangular part of the correlation matrix. We call this distance the clock correlation distance (CCD). A smaller CCD indicates greater similarity in clock gene co-expression. Although here we used Euclidean distance, other distance metrics could be used as well.

To estimate the probability that the observed CCD is less than would be expected by chance, we randomly selected 12 genes (from all genes measured in that dataset) 1,000 times, in order to estimate a null distribution of CCDs. We then calculated the exact one-sided *p*-value using the number of random samplings that gave a CCD less than or equal to the observed CCD and the method of Phipson and Smyth in the statmod R package (*Phipson & Smyth, 2010*). For the eight mouse datasets that form the reference, we calculated the CCD between each unique pair of datasets, essentially taking one dataset in each pair to be the reference.

To quantify the extent to which clock gene co-expression is perturbed in samples from one condition (e.g., tumor) relative to samples from another condition (e.g., non-tumor) in a given dataset, we calculated the CCD for each condition relative to the mouse reference, then calculated the difference between those CCDs. We call this difference the delta clock correlation distance (ΔCCD). In this study, a positive ΔCCD indicates that clock gene co-expression in tumor (or knockout) samples is less similar to the reference than is clock gene co-expression in non-tumor (or wild-type) samples.

To evaluate the probability that the ΔCCD is greater than would be expected by chance, we permuted the condition labels of the samples and recalculated the ΔCCD 1,000 times, always keeping the reference fixed. We then calculated the exact one-sided *p*-value using the number of permutations that gave a ΔCCD greater than or equal to the observed ΔCCD

and the method of Phipson and Smyth in the statmod R package (*Phipson & Smyth, 2010*). Since we used the one-sided *p*-value, the alternative hypothesis was that non-tumor (or wild-type) is more similar to the reference than is tumor (or knockout).

To calculate the ΔCCD for individual tumor grades in TCGA data, we used the clinical metadata provided in GSE62944. We analyzed all combinations of TGCA cancer type and tumor grade that included at least 50 tumor samples. In each case, we calculated the ΔCCD using all non-tumor samples of the respective cancer type.

To compare ΔCCD and tumor purity, we used published consensus purity estimates for TCGA tumor samples (*Aran, Sirota & Butte, 2015*). The estimates are based on DNA methylation, somatic copy number variation, and the expression of immune genes and stromal genes (none of which are clock genes).

To quantify differential expression between tumor and non-tumor samples and between knockout and wild-type samples, we used limma and voom (*Smyth, 2004*; *Law et al., 2014*). To ensure a fair comparison between human and mouse data, we ignored time of day information in the mouse samples. We quantified the variation in expression of clock genes in each dataset and condition using the median absolute deviation (MAD), which is less sensitive to outliers than the standard deviation.

## Analyzing circadian gene expression in human lung cancer

We obtained the set of rhythmic transcripts (identified by microarray probe ID) inferred by CYCLOPS to be rhythmic in healthy human lung (*Anafi et al., 2017*). This set contains transcripts whose abundance was well described by a sinusoidal function of CYCLOPS phase in samples from both the Laval and GRNG sites of GSE23546 (Bonferroni-corrected $P < 0.05$ and peak/trough ratio >2) and whose orthologs were rhythmic in mouse lung. We mapped the microarray probes to Entrez Gene IDs and calculated the acrophase for each gene as the circular mean of the provided acrophase for the corresponding probes. Phase values inferred by CYCLOPS are relative, so Anafi et al. adopted the convention of setting phase $\pi$ (in radians) to the average acrophase of the PAR bZip transcription factors (DBP, HLF, and TEF) (*Anafi et al., 2017*).

Because some of the module preservation statistics calculated by NetRep use the expression matrix directly, we used ComBat (*Johnson, Li & Rabinovic, 2007*) to merge the expression data from the Laval and Groningen sites of GSE23546. We then followed WGCNA's recommended procedure for identifying gene modules, i.e., groups of genes with correlated expression across samples (*Langfelder & Horvath, 2008*). Briefly, starting with the merged expression data, we calculated the Spearman correlation matrix for the 1,292 genes, then used the signed method with a soft thresholding power of 12 to calculate the adjacency matrix, then calculated the topological overlap dissimilarity matrix. We used hierarchical clustering of the dissimilarity matrix and adaptive branch pruning to define modules of genes, followed by a procedure to merge closely related modules. We then used the DAVID web application (version 6.8) to discover functional categories enriched in each module (*Huang, Sherman & Lempicki, 2009*).

We used NetRep (*Ritchie et al., 2016*) to calculate seven module preservation statistics between healthy human lung and the non-tumor and tumor samples from five datasets

of human lung cancer (LUAD and LUSC from TCGA and GSE19188, GSE19804, and GSE32863 from NCBI GEO). We excluded GSE10072, because it included expression levels for only 876 of the 1,292 genes (the other five datasets included at least 1,199 of the 1,292 genes). When calculating module preservation, NetRep automatically removed any genes that were not measured in both healthy human lung and the group of samples to which it was being compared. For non-tumor and tumor samples from each dataset, we calculated the Spearman correlation matrix and adjacency matrix using the same procedure as for healthy human lung.

To evaluate the statistical significance of the difference in each module preservation statistic between non-tumor and tumor samples, we performed permutation testing and calculated one-sided $p$-values similarly to the procedure for $\Delta$CCD, permuting the sample labels (non-tumor or tumor) in the test dataset and recalculating module preservation 1,000 times. Note this is different from the standard way WGCNA and NetRep perform permutations, which is to shuffle the gene labels.

Periodic smoothing splines of $\log_2$ fold-change as a function of acrophase in healthy lung were fit using ZeitZeiger (*Hughey, Hastie & Butte, 2016*). *P*-values of non-zero amplitude of the spline fits were calculated using 1,000 permutations of the relationship between acrophase and $\log_2$ fold-change and the method of Phipson and Smyth (*Phipson & Smyth, 2010*). For Phase Set Enrichment Analysis (*Zhang et al., 2016*), we used the canonical pathways gene sets from MSigDB (*Liberzon et al., 2011*) and looked for gene sets with a $q$-value $\leq 0.1$ against a uniform distribution and vector-average value within 0.4 radians of either 0.07 $\pi$ or 1.06 $\pi$ (the mean phases of the peak and trough of the spline fits, respectively). No gene sets met the criteria for the former.

## RESULTS

### Consistent co-expression of clock genes in mice

The progression of the mammalian circadian clock is marked by daily rhythms in gene expression throughout the body (*Zhang et al., 2014*). We hypothesized that the relative phasing of different genes' rhythms would result in those genes having a characteristic pattern of co-expression. Such a pattern could be used to infer the progression of the clock within a group of samples, even if the samples are not labeled with time of day, as long as they were taken from multiple parts of the circadian cycle.

Although most circadian gene expression is tissue-specific, the rhythms of a select group of genes show consistent relative phasing in multiple organs in mice and humans (*Hughey & Butte, 2016*). We focused on 12 genes from this group, each of which is part of the core clock or directly controlled by the clock, in order to define a general signature of circadian clock progression. For the rest of the paper, we refer to these 12 genes as "clock genes."

As a first test of our hypothesis, we assembled eight publicly available datasets of genome-wide circadian gene expression from various healthy mouse organs (*Oster et al., 2006*; *Hoogerwerf et al., 2008*; *Hughes et al., 2009*; *Negoro et al., 2012*; *Geyfman et al., 2012*; *Haspel et al., 2014*; *Zhang et al., 2014*) (Table S1). The organs were collected from mice under both constant darkness and alternating light-dark cycles. For each dataset, we

calculated the Spearman correlation between expression values (over all samples) of each pair of clock genes.

The pattern of co-expression was highly consistent across datasets (for 59 of 66 gene pairs, the sign of the Spearman correlation was the same in at least six of eight datasets; Fig. S1). Co-expression revealed two groups of genes, where the genes within a group tended to be positively correlated with each other and negatively correlated with genes in the other group. Genes in the first group (Arntl, Npas2, and Clock), which form the positive arm of the clock (*Partch, Green & Takahashi, 2014*), peaked in expression shortly before zeitgeber time 0 (ZT0, which corresponds to time of lights on or sunrise; Fig. S2). Genes in the second group (Cry2, Nr1d1, Nr1d2, Per1, Per2, Per3, Dbp, and Tef), which form the negative arms of the clock, peaked in expression near ZT10. Cry1, which was part of the first group in some datasets and the second group in others, tended to peak in expression near ZT18. These results indicate that the progression of the circadian clock in various mouse organs produces a consistent pattern of co-expression between clock genes.

To construct a single reference pattern of clock gene co-expression, we combined the eight datasets in a fixed-effects meta-analysis (Figs. 1A–1B and Materials and Methods). To compare clock gene co-expression between the reference and a group of samples from another dataset, we developed a metric we call the clock correlation distance (CCD), which corresponds to the Euclidean distance between the respective correlation vectors (Fig. 1A). A smaller CCD indicates a greater similarity in clock gene co-expression. To estimate the probability that the observed CCD is less than would be expected by chance, we estimated the distribution of CCD that would result from using 12 randomly selected genes instead of the 12 clock genes. As a positive control, we calculated the Euclidean distance between the correlation vectors corresponding to all pairs of the eight mouse datasets ($P \leq 0.001$ by permutation test for each pair; Fig. 1C and Fig. S3).

We used the reference pattern and the CCD metric to verify that clock gene co-expression arises from the phasing of the genes' rhythms relative to each other, not their phasing relative to time of day. Thus, the pattern is not affected by phase differences between groups of samples, such as those caused by daytime feeding in mice (*Vollmers et al., 2009*) ($P \leq 0.001$ for CCD relative to reference; Fig. S4).

Most computational methods for quantifying circadian rhythmicity and inferring the status of the clock require that samples be acquired over the entire 24-h cycle. Because our approach does not attempt to infer oscillations, we wondered if it would be robust to partial coverage of the 24-h cycle. We therefore analyzed clock gene co-expression in three of the original eight datasets, in samples acquired during the first 8 h of the day (or subjective day) or the first 8 h of the night (or subjective night). In each dataset, clock gene co-expression was preserved in both daytime and nighttime samples ($P \leq 0.001$; Fig. S5). Thus, our approach can detect the signature of clock progression in groups of samples without using time of day information, even if the samples' coverage of the 24-h cycle is incomplete.

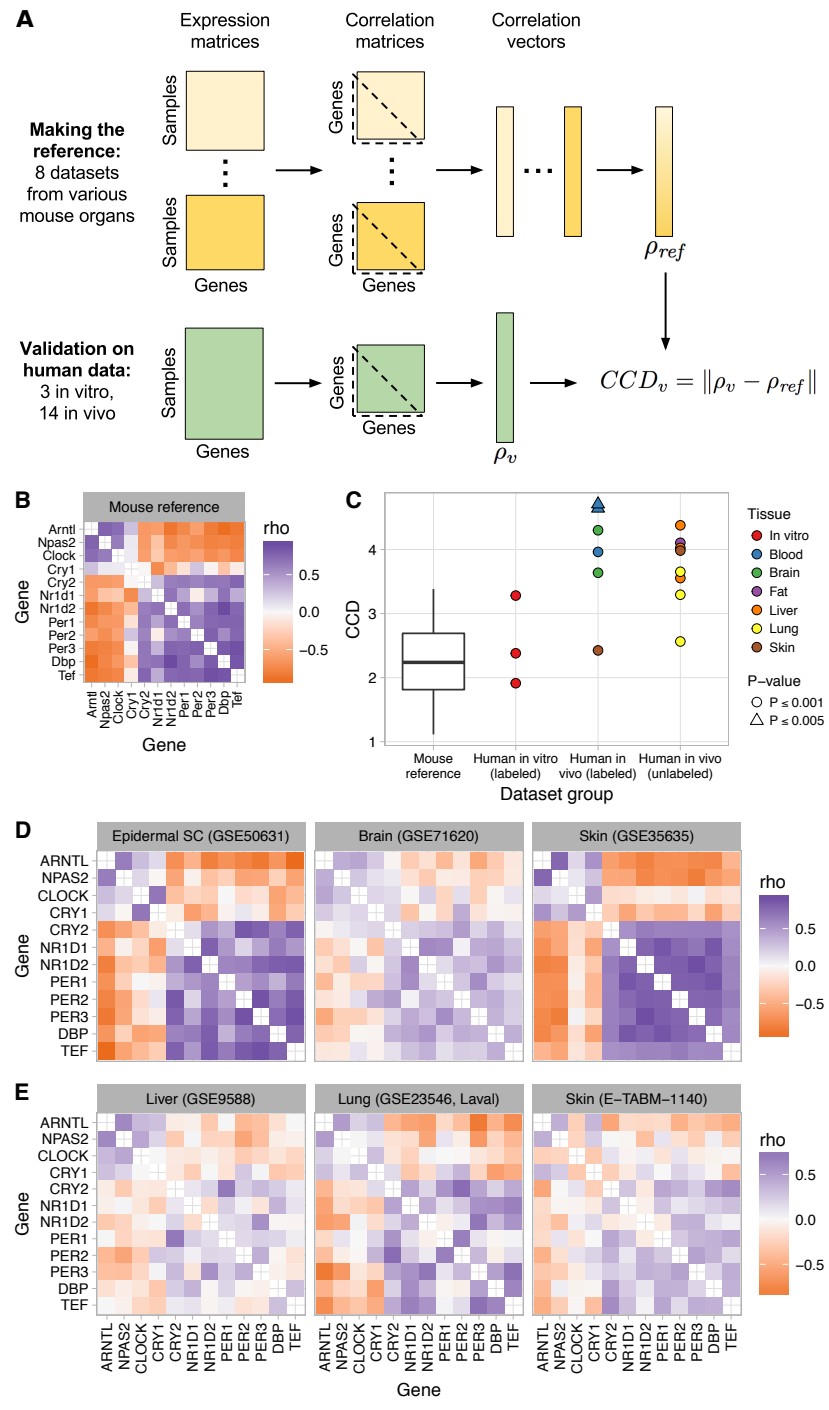

**Figure 1** **Consistent patterns of clock gene co-expression in mice and humans.** (A) Schematic of procedure for constructing a reference pattern of clock gene co-expression from healthy mouse organs and for comparing clock gene co-expression in an independent dataset to the mouse reference using the clock correlation distance (CCD). (B) Heatmap of Spearman correlation (rho) between clock genes for the mouse reference, based on a fixed-effects meta-analysis as described in the 

**Figure 1 (…continued)**
Materials and Methods. (C) Clock correlation distance (CCD) for mouse and human datasets. For the boxplot corresponding to the mouse reference, distances were calculated between each pair of datasets (8 datasets give 28 unique pairs; Fig. S3). For human datasets, the CCD was calculated relative to the mouse reference. Example human (labeled) datasets are shown in (D), and example human (unlabeled) datasets are shown in (E). *P*-value corresponds to the probability that 12 randomly selected genes (instead of the 12 clock genes) could produce a CCD less than or equal to the one observed. (D) Heatmaps of Spearman correlation in three human datasets for which samples are labeled with time of day (or time since synchronization, for GSE50631). (E) Heatmaps for three datasets not designed to study circadian rhythms and for which samples are not labeled with time of day. Heatmaps not shown here are shown in Fig. S6. All heatmaps of clock gene co-expression in Figs. 1 and 2, and the supplemental figures have the same mapping of correlation value to color, so they are directly visually comparable.

## Validation of the co-expression pattern in human tissues

We next examined clock gene co-expression in nine publicly available datasets of circadian gene expression from human tissues: one from skin, two from brain, three from blood, and three from cells cultured *in vitro* (*Hughes et al., 2009*; *Spörl et al., 2012*; *Möller-Levet et al., 2013*; *Li et al., 2013*; *Janich et al., 2013*; *Archer et al., 2014*; *Hoffmann et al., 2014*; *Arnardottir et al., 2014*; *Chen et al., 2016*) (Table S1). Samples from these datasets are labeled with time of day (or in the case of cells *in vitro*, time since synchronization) and were taken from multiple time-points in the circadian cycle. Other than the two datasets of U2OS cells, which are derived from an osteosarcoma but show normal rhythms of the core clock genes (*Vollmers, Panda & DiTacchio, 2008*; *Maier et al., 2009*), all samples were from healthy tissue.

Although gene expression rhythms in mice and humans show different phasing relative to the light-dark cycle (*Hughey & Butte, 2016*), the pattern of clock gene co-expression in the human datasets significantly resembled the reference pattern from mice ($P \leq 0.001$ for three of three *in vitro* datasets and four of six *in vivo* datasets; Figs. 1C–1D and Fig. S6). The CCDs of the datasets varied: skin and *in vitro* cells had the smallest (i.e., the strongest similarity to mice), brain was intermediate, and blood had the largest (Fig. 1C). Correlations between clock genes in the latter two organs were generally weaker than those in mice (Fig. S7), consistent with noisier circadian rhythms in those datasets (*Hughey & Butte, 2016*). The co-expression pattern in the human skin dataset, for which samples were acquired from each individual at three time-points (9:30 am, 2:30 pm, and 7:30 pm), was due to co-expression both across individuals at each time-point and across time-points within each individual (Fig. S8).

To confirm our findings in additional human organs, we analyzed eight transcriptome datasets from healthy human lung, liver, skin, and adipose tissue (*Schadt et al., 2008*; *Innocenti et al., 2011*; *Bossé et al., 2012*; *Grundberg et al., 2012*; *Bonder et al., 2014*) (Table S1). Samples from these datasets were not collected for the purpose of studying circadian rhythms. Thus, the samples are not labeled with time of day and may not cover the entire circadian cycle. Nonetheless, clock gene co-expression in these datasets was similar to that of mice ($P \leq 0.001$ for all 8 datasets; Figs. 1C, 1E, and Fig. S9), with CCDs comparable to those of datasets designed to quantify circadian rhythms in humans *in vivo*. The generally weaker co-expression patterns and higher CCDs in human *in vivo* datasets compared

to mouse datasets is likely due to data collected from humans having more sources of variation (including technical, environmental, and genetic variation). We conclude that our approach can detect the signature of circadian clock progression in publicly available transcriptome data from various human tissues, even in datasets not designed to quantify circadian rhythms.

## Aberrant patterns of clock gene co-expression in human cancer

We then turned our attention to clock gene co-expression in human cancer. We started with RNA-seq data collected by The Cancer Genome Atlas (TCGA) and reprocessed using the Rsubread package (*Rahman et al., 2015*). TCGA samples are from surgical resections performed prior to neoadjuvant treatment. We analyzed TCGA data from the 12 cancer types that included at least 30 samples from adjacent non-tumor tissue (Table S1). To study human hepatocellular carcinoma and non-small cell lung cancer in more depth, we also analyzed eight datasets from the NCBI Gene Expression Omnibus (GEO), each of which had tumor and adjacent non-tumor samples (*Landi et al., 2008*; *Hou et al., 2010*; *Lu et al., 2010*; *Roessler et al., 2010*; *Lamb et al., 2011*; *Selamat et al., 2012*; *Lim et al., 2013*; *Villa et al., 2016*) (Table S1). For all data from TCGA and GEO, the times of day of surgery are not available. Presumably, most to all of the samples were acquired during surgical working hours (6:00 am to 6:00 pm).

To quantify the extent to which clock gene co-expression is perturbed in tumor samples compared to non-tumor samples in a given dataset, we calculated the difference between the CCDs (relative to the mouse reference) for each condition. We call this difference the delta clock correlation distance ($\Delta$CCD), where a $\Delta$CCD >0 indicates that clock gene co-expression in tumor samples is less similar to the reference than that of non-tumor samples (Fig. 2A). We continued to use the reference from mouse data, instead of one based on human data, in order to maintain a strict separation of training and validation data and because in the mouse data, the effects of other sources of variation and confounding are minimized. Thus, the mouse data offered the clearest window into the clock gene co-expression that results from circadian clock progression *in vivo*. To evaluate the probability that the $\Delta$CCD is greater than would be expected by chance, we permuted the condition labels of the samples and recalculated the $\Delta$CCD 1,000 times.

Clock gene co-expression in non-tumor samples was similar to that observed in solid healthy human tissues ($P \leq 0.005$ for CCD relative to the mouse reference in 14 of 20 datasets; Figs. 2B–2C and Fig. S10). In contrast, co-expression in tumor samples often lacked a significant similarity to the expected pattern ($P > 0.05$ for CCD in 10 of 20 datasets; Figs. 2B–2C and Fig. S10). Consequently, tumor CCDs were significantly higher than non-tumor CCDs ($P = 1.4 \cdot 10^{-8}$ by paired $t$-test) and each human cancer dataset had a $\Delta$CCD >0 ($P \leq 0.005$ in 15 of 20 datasets; Fig. 2D and Fig. S11A). Among the three TCGA cancer types with the lowest $\Delta$CCD, prostate adenocarcinoma had a relatively high non-tumor CCD, whereas renal clear cell carcinoma and thyroid carcinoma each had a relatively low tumor CCD (Fig. S11B). Interestingly, we observed no clear trend between $\Delta$CCD and histological tumor grade (Fig. S12). Overall, these results suggest that circadian clock progression is perturbed in a range of human cancers.

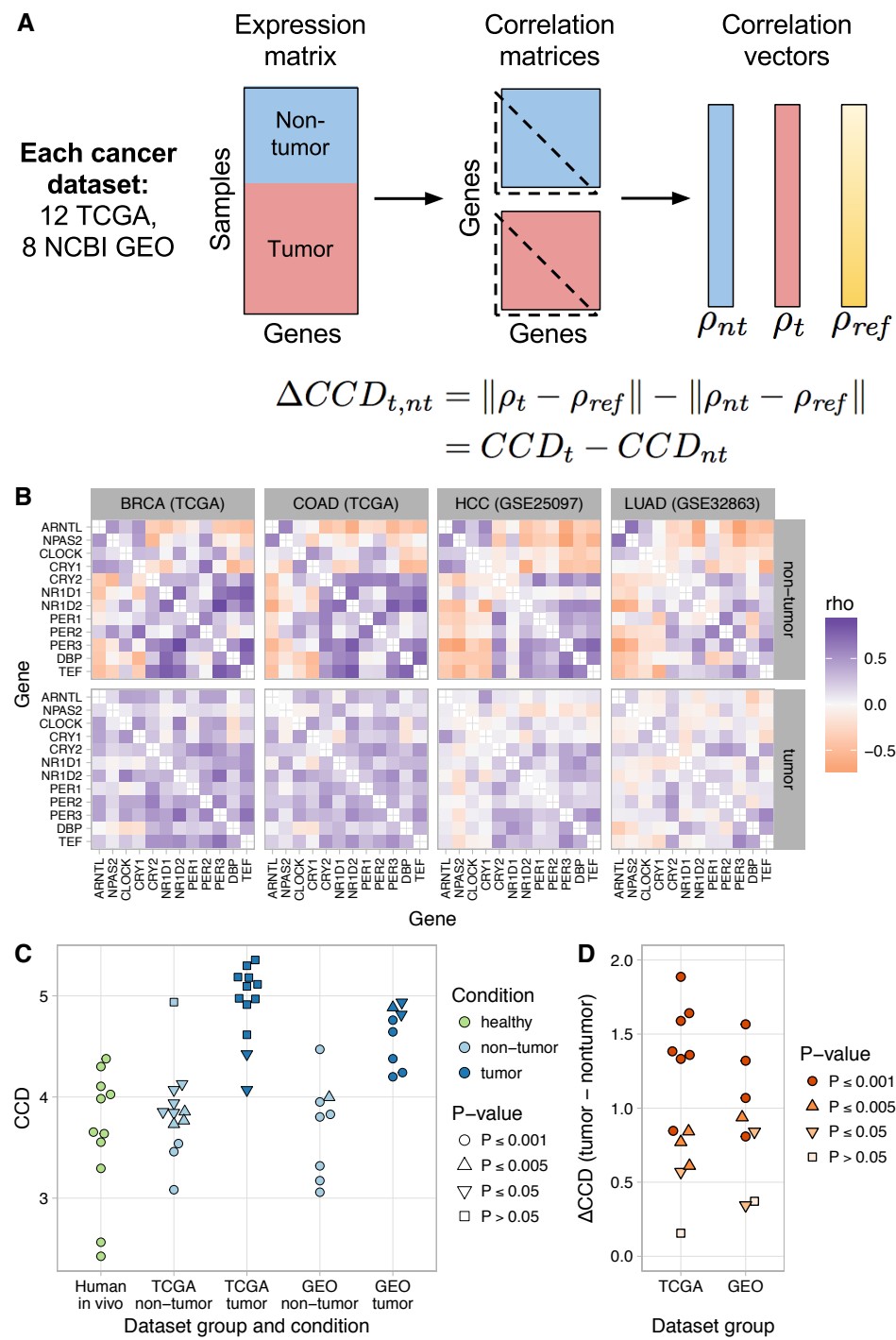

Figure 2 **Loss of normal clock gene co-expression in human tumor samples from various cancer types.**
(A) Schematic of procedure for comparing clock gene co-expression, relative to the mouse reference, between non-tumor and tumor samples from human cancer datasets. (B) Heatmaps of Spearman correlation between clock genes for non-tumor and tumor samples from two (continued on next page...)

**Figure 2 (...continued)**
TCGA cancer types and two NCBI GEO datasets. Abbreviations: breast invasive cell carcinoma (BRCA), colon adenocarcinoma (COAD), hepatocellular carcinoma (HCC), lung adenocarcinoma (LUAD). Heatmaps for the other 16 human cancer datasets are shown in Fig. S9. (C) Clock correlation distance (CCD) for non-tumor and tumor samples relative to the mouse reference. Each point corresponds to one condition in one dataset. "Human *in vivo*" corresponds to the human *in vivo* datasets shown in Fig. 1C (labeled and unlabeled), minus the human blood datasets. *P*-value corresponds to the probability that 12 randomly selected genes (instead of the 12 clock genes) could produce a CCD less than or equal to the one observed. (D) Delta clock correlation distance ( ΔCCD) between non-tumor and tumor samples in 12 TCGA cancer types and 8 datasets from NCBI GEO. Each point corresponds to one dataset. Positive ΔCCD indicates perturbed clock gene co-expression in tumor samples relative to non-tumor samples. *P*-value corresponds to the probability that a random permutation of the samples' condition labels could produce a ΔCCD greater than or equal to the one observed.

Tumors are a complex mixture of cancer cells and various non-cancerous cell types. The proportion of cancer cells in a tumor sample is called the tumor purity and is an important factor to consider in genomic analyses of bulk tumors (*Aran, Sirota & Butte, 2015*). We therefore examined the relationship between ΔCCD and tumor purity in the TCGA data. With the exception of thyroid carcinoma and prostate adenocarcinoma, ΔCCD and median tumor purity across TCGA cancer types were positively correlated (Fig. S13; Spearman correlation = 0.67, $P = 0.059$ by exact test). These findings suggest that at least in some tumors, clock progression is perturbed more strongly in cancer cells than in non-cancerous cells.

## Distinct patterns of clock gene expression in human cancer and clock gene knockouts in mice

To better understand the nature of the dysregulation of clock progression, in particular to determine if the pattern of clock gene co-expression in human tumors could be explained by inactivation of core clock genes, we compared clock gene expression in human cancer and in clock gene knockouts in mice. We assembled seven datasets of circadian gene expression, measured in various tissues, that included samples from wild-type mice and from mice in which at least one core clock gene was knocked out, either in the entire animal or in a specific cell type (*Vollmers et al., 2009*; *Cho et al., 2012*; *Nikolaeva et al., 2012*; *Paschos et al., 2012*; *Dyar et al., 2014*; *Young et al., 2014*; *Dudek et al., 2016*) (Table S1). For each dataset, we calculated clock gene co-expression in wild-type and knockout samples (Fig. S14).

The two datasets with the highest ΔCCD (>50% higher than any ΔCCD we observed in human cancer) were those in which the knockout mice lacked not one, but two components of the clock (Cry1 and Cry2 in GSE13093; Nr1d1 and Nr1d2 in GSE34018; Fig. 3A). The ΔCCDs for the other five datasets were similar to or somewhat lower than the ΔCCDs we observed in human cancer. Due to the smaller sample sizes compared to the human cancer datasets, the ΔCCDs for four of seven datasets of clock gene knockouts were not significantly greater than zero ($P > 0.05$).

To further compare clock gene expression in human cancer and clock gene knockouts, we calculated differential expression of the clock genes between non-tumor and tumor samples and between wild-type and knockout samples (Fig. 3B). Differential expression in the knockouts was largely consistent with current understanding of the core clock. For

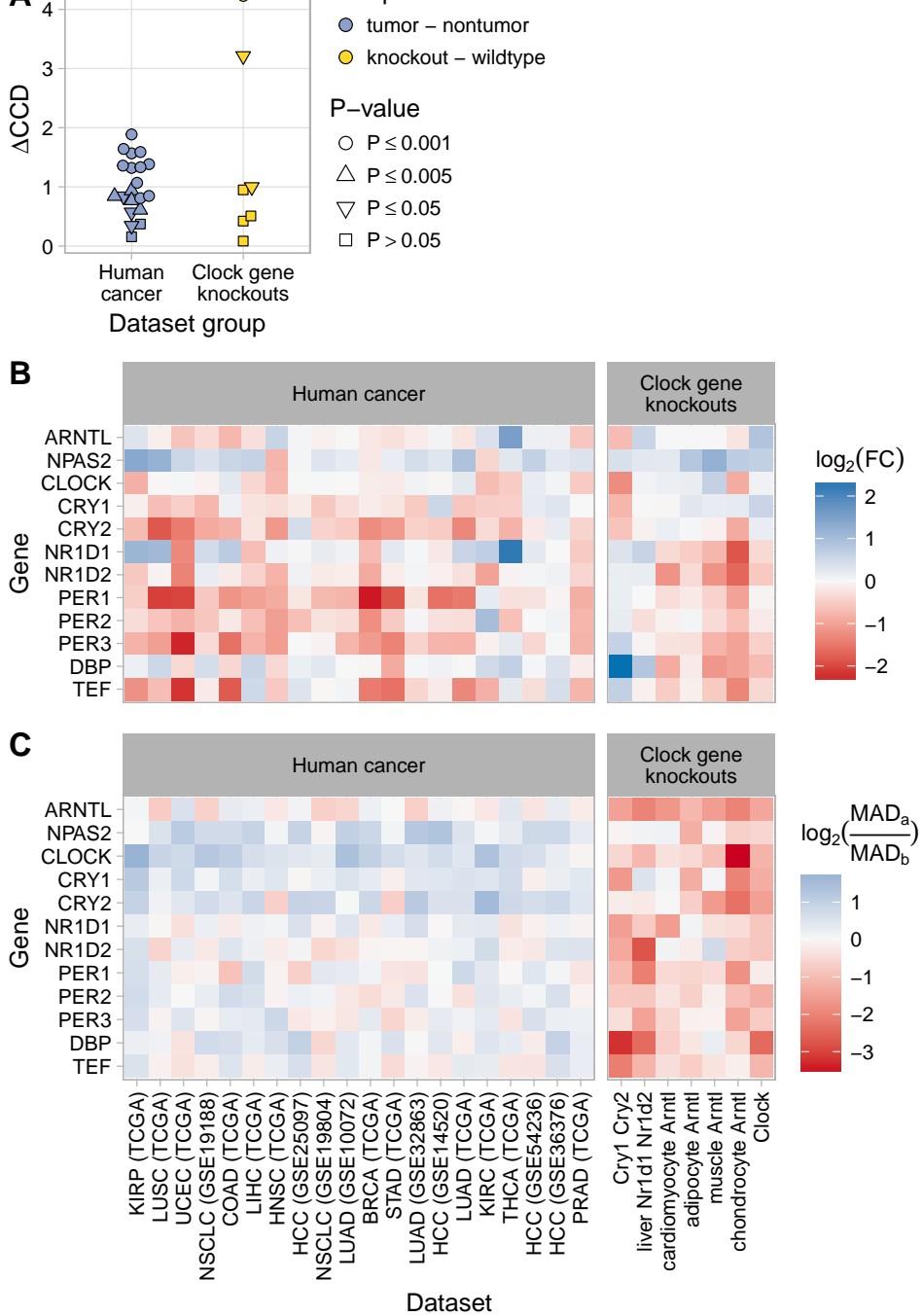

**Figure 3 Changes in clock gene expression in human cancer are distinct from those caused by knock-out of the clock genes in mice.** (A) Delta clock correlation distance (ΔCCD) for human cancer datasets (TCGA and GEO) and for clock gene knockout datasets from mice. (B) Heatmaps of the estimated log₂ fold-change in expression between tumor and non-tumor samples or between knockout and wild-type samples. A positive value indicates higher expression in tumor samples (continued on next page...)

**Figure 3 (...continued)**
or knockout samples, respectively. (C) Heatmaps of the $\log_2$ ratio of the median absolute deviation (MAD) of expression in tumor compared to non-tumor samples and knockout compared to wild-type samples. In the legend, $MAD_a$ refers to tumor or knockout samples, $MAD_b$ refers to non-tumor or wild-type samples. A positive value indicates that the variation in expression of that gene is greater in tumor (or knockout) samples than in non-tumor (or wild-type) samples. For RNA-seq data (TCGA and chondrocyte Arntl KO), expression values were based on $\log_2$ (TPM + 1), where TPM is transcripts per million. For microarray data, expression values were based on log-transformed, normalized intensity. Within each dataset group, datasets are ordered by descending ΔCCD.

example, knockout of Arntl (the primary transcriptional activator) tended to cause reduced expression of Nr1d1, Nr1d2, Per1, Per2, Per3, Dbp, and Tef, and increased or unchanged expression of the other clock genes. In the double knockout of Cry1 and Cry2 (two negative regulators of the clock), this pattern was reversed. Interestingly, neither of these patterns of differential expression was apparent in human cancer.

In the clock gene knockouts, rhythmic expression of most clock genes was reduced or lost (calculated using each sample's time of day; Fig. S15). Without time of day information in the human cancer datasets, it was not possible to directly quantify rhythms of clock genes in non-tumor and tumor samples. However, we reasoned that a proxy for rhythmicity could be the magnitude of variation in expression. Therefore, for each human cancer dataset, we calculated the median absolute deviation (MAD) in expression of the clock genes in non-tumor and tumor samples. We then compared the $\log_2$ ratios of MAD between tumor and non-tumor samples to the $\log_2$ ratios of MAD between knockout and wild-type samples from the clock gene knockout data (Fig. 3C). As expected, samples from clock gene knockouts showed widespread reductions in MAD compared to samples from wild-type mice. In contrast, human tumor samples tended to show similar or somewhat higher MAD compared to non-tumor samples. The distinct patterns of differential expression and differential variability between tumor and non-tumor samples, compared to knockout and wild-type samples, imply that dysregulation of clock progression in human cancer is not due solely to the inactivation of one or more clock genes.

## Dysregulation of broader circadian gene expression in human lung cancer

Finally, we explored how dysregulation of clock progression might affect the circadian transcriptome. We obtained a set of 1,292 genes that were inferred by the CYCLOPS method to be rhythmic in a dataset from healthy human lung (*Anafi et al., 2017*). By applying WGCNA (*Langfelder & Horvath, 2008*) to the same dataset, we grouped the 1,292 genes into five modules according to their co-expression (Fig. 4A). Based on DAVID (*Huang, Sherman & Lempicki, 2009*) and consistent with the analysis of *Anafi et al. (2017)*, the five modules were enriched for genes involved in various biological processes, including angiogenesis, phosphoprotein signaling, and alternative splicing (Table S3). We then used NetRep (*Ritchie et al., 2016*) to determine the extent to which each module was preserved in non-tumor and tumor samples from five datasets of human lung cancer (two from TCGA and three from NCBI GEO). Notably, all five datasets had ≤65 non-tumor samples and the three datasets from NCBI GEO had ≤91 tumor samples, too few for CYCLOPS to produce

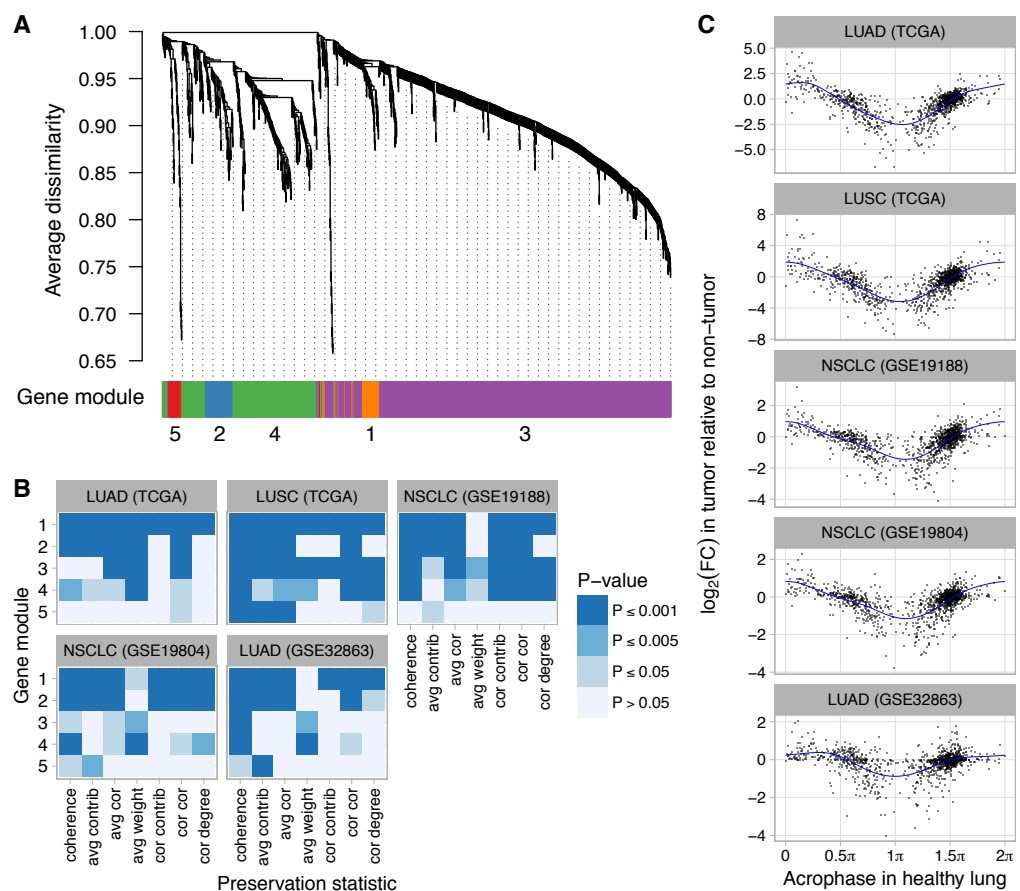

**Figure 4 Perturbed expression of normally rhythmic genes in human lung cancer.** (A) Hierarchical clustering of 1,292 genes inferred to have a circadian rhythm in healthy human lung. Gene modules were defined by following the procedure recommended by WGCNA. The number of genes in modules 1–5 are 62, 75, 828, 291, and 36. (B) Statistical significance of differential module preservation between non-tumor and tumor samples for seven preservation statistics for each module in each dataset. One-sided *p*-values are based on 1,000 permutations of the sample labels in the respective test dataset. Abbreviations: lung adenocarcinoma (LUAD), lung squamous cell carcinoma (LUSC), non-small cell lung cancer (NSCLC). (C) Scatterplots of $\log_2$ fold-change in each lung cancer dataset vs. the phase of peak expression (acrophase, in radians) in healthy lung. Acrophase $\pi$ is defined to be the circular mean of the acrophases of the circadian clock-driven PAR bZip transcription factors DBP, TEF, and HLF. Each point corresponds to one of the 1,292 genes. Each blue curve indicates a fit to a periodic smoothing spline. The circular mean of the trough of the spline fits is $1.06\pi$.

a reliable ordering. Based on this analysis, four of five modules (comprising 1,256 of 1,292 rhythmic genes) were significantly more strongly preserved in non-tumor compared to tumor samples (Fig. 4B, $P \leq 0.001$ for at least two of seven preservation statistics in at least four of five datasets).

We also quantified differential expression of the 1,292 normally rhythmic genes between non-tumor and tumor samples in the five lung cancer datasets. We then examined the relationship between each gene's $\log_2$ fold-change (tumor vs. non-tumor) and its phase

of peak expression (acrophase) in healthy lung as inferred by CYCLOPS. Remarkably, we observed a consistent pattern, in which genes with an acrophase near $\pi$ tended to show the strongest decrease in expression in tumor samples (Fig. 4C, $P \leq 0.001$ of non-zero amplitude by permutation testing for each dataset). Using Phase Set Enrichment Analysis (*Zhang et al., 2016*), we found that genes with an acrophase near $\pi$ were particularly enriched for involvement in G protein-coupled receptor signaling (Table S4). Taken together, these findings suggest that perturbed clock progression in human lung cancer is accompanied by systematic changes in broader circadian gene expression.

## DISCUSSION

Increasing evidence has suggested that systemic disruption of the circadian clock can promote tumor development and that components of a tumor can disrupt the circadian clock (reviewed in *Lamia, 2017*). Until now, however, whether the clock is progressing, i.e., oscillating, normally within human tumors has been unclear. Here we developed a simple method to detect the signature of clock progression based on the co-expression of a small set of genes. By applying the method to publicly available cancer transcriptome data, we discovered that clock progression may be perturbed in multiple types of human cancer. Our analysis also suggests that dysregulation of clock progression in cancer is not caused solely by inactivation of core clock genes and is accompanied by large-scale changes in circadian gene expression and co-expression.

Our approach relies on three principles. First, we use prior knowledge of clock genes and clock-controlled genes. Second, we account for the fact that clock progression is defined not by a static condition, but by a dynamic cycle. Our approach thus exploits the co-expression of clock genes that arises from (1) different genes having rhythms with different circadian phases and (2) different samples being taken from different points in the cycle. Finally, our method does not attempt to infer an oscillatory pattern, but instead uses only the statistical correlations in expression between pairs of genes. The underlying assumption of the $\Delta$CCD is that perturbations to the clock will alter the relative phases and/or signal-to-noise ratios of rhythms in clock gene expression and thereby alter the correlations. Although the correlation matrix captures only part of the complex relationship between genes, it is intuitive, simple to calculate, and requires relatively few samples (our results indicate that 30 in humans may be sufficient). Altogether, these principles enable our method to detect perturbed clock progression in groups of samples whose times of acquisition are unknown and whose coverage of the 24-h cycle is incomplete. Consequently, the CCD and $\Delta$CCD should be a valuable complement to methods designed to infer rhythms in omics data, such as CYCLOPS (*Anafi et al., 2017*).

Despite these advantages, our method does have limitations. First, a large CCD does not exclude the possibility that the clock genes are still rhythmic. Instead, it implies that if rhythms are present, the phase relationships between genes are greatly altered relative to a normally progressing clock. Similarly, despite altered expression and co-expression levels, some of the genes rhythmic in healthy lung may still be rhythmic in lung cancer. Second, clock gene co-expression is insensitive to the alignment of the circadian clock

to time of day, and so cannot detect a phase difference between conditions, such as that recently observed during manic episodes of bipolar disorder (*Moon et al., 2016*). This limitation, however, allowed us to readily compare clock gene co-expression between mice and humans, despite the circadian phase difference between the two species (*Hughey & Butte, 2016*). Third, as shown by our analysis of clock gene knockouts in mice, larger sample sizes are required to find differences in co-expression than to find differences in rhythms (the former does not require time of day information, the latter does). This is partly because differences in co-expression are invariant to the relative levels of gene expression between conditions, which is why we also analyzed differential expression and differential variability. Fourth, transcription is only one facet of the core clock mechanism, and perturbations to post-translational modification or degradation of clock proteins (if unaccompanied by changes in clock gene expression) would not be detected by our approach. Finally, because co-expression is calculated for a group of samples, our approach does not immediately lend itself to assessing clock progression in single samples. Thus, although our results support dysregulated clock progression in tumors as a group, there may be some tumors in which the clock is progressing normally. In the future, it may be possible to complement the $\Delta$CCD metric and infer clock progression in a given individual by directly comparing matched healthy and disease samples.

In healthy tissues *in vivo*, the circadian clocks of individual cells are entrained and oscillating together, which allows bulk measurements to contain robust circadian signals. Consequently, the loss of a circadian signature in human tumors could result from dysfunction in either entrainment, the oscillator, or both. Dysfunction in entrainment would imply that the clocks in at least some of the cancer cells are out of sync with each other and therefore free running, i.e., ignoring zeitgeber signals. Dysfunction in the oscillator would imply that the clocks in at least some of the cancer cells are no longer progressing normally. Given the current data, in which expression values are the averages across many cells, these scenarios cannot be distinguished. However, the moderate correlation between $\Delta$CCD and tumor purity across cancer types leads us to speculate that the clocks in stromal and/or infiltrating immune cells may be progressing normally. Notably, if clock gene co-expression in tumors were normal, then the moderate increases in expression variability of clock genes could imply rhythms (synchronized across the tumor) of increased amplitude. Because clock gene co-expression in tumors is strongly perturbed, however, we find this scenario unlikely. We find it more likely that increased gene expression variability in tumors is a result of non-circadian sources of variation, e.g., tumor grade and subtype. In the future, the specific mechanisms of perturbed clock progression may be unraveled through a combination of mathematical modeling (*Kim & Forger, 2012*; *Lück et al., 2014*) and single-cell measurements. A separate matter not addressed here is how cancer influences circadian rhythms in the rest of the body (*Masri et al., 2016*), which may be relevant for optimizing the daily timing of anticancer treatments (*Dulong et al., 2015*).

An apparent contradiction to perturbed clock progression in tumors is the presence of normal rhythms in U2OS cells (which are derived from an osteosarcoma) and some other cancer cell lines (*Relógio et al., 2014*), as well as in human and mouse glioblastoma

cells *in vitro* (*Slat et al., 2017*). One explanation for this contradiction is that there could be a critical difference, in terms of circadian clock progression, between tumors *in vivo* and cancer cells *in vitro*. If so, we would speculate that the dysregulation of clock progression in some human tumors may, under certain conditions, be reversible.

Based on this study alone, which is observational, it is not possible to determine the direction of causality between tumor development and clock dysregulation, or to determine whether clock dysregulation provides a selective advantage. Indeed, perturbations to co-expression in cancer are not limited to clock genes or to rhythmic genes, as previous work has found changes in co-expression and connectivity across the transcriptome (*West et al., 2012*; *Anglani et al., 2014*). However, given the clock's established role in regulating metabolism and a recent finding that stimulation of the clock inhibits tumor growth in melanoma (*Kiessling et al., 2017*), our findings raise the possibility that clock dysregulation and manipulation of normal circadian physiology may be a cancer driver in multiple solid tissues. On the other hand, a functional circadian clock seems to be required for growth of acute myeloid leukemia cells (*Puram et al., 2016*), so further work is necessary to clarify this issue.

## CONCLUSIONS

Although the effect of restoring normal clock progression remains to be tested at the pre-clinical and clinical stages, our findings raise the conjecture that this could be a viable treatment strategy in a wide range of cancer types. Recent work on the effects of natural light exposure (*Stothard et al., 2017*) and time-restricted feeding (*Hatori et al., 2012*; *Chaix et al., 2014*) suggests that such a treatment strategy would not necessarily have to be pharmacological. Finally, given the practical challenges of studying circadian rhythms at the cellular level in humans, our method offers the possibility to infer clock progression in diverse human phenotypes using publicly available transcriptome data.

## ACKNOWLEDGEMENTS

We thank Dvir Aran and the VUMC Editor's Club for helpful comments on the manuscript.

### Funding

This work was supported by NIH R35GM124685 to Jacob J. Hughey, by NIH 1U2COD023196 and U01HG009086 to Guanhua Chen, and by the SyBBURE Searle Undergraduate Research Program to Jarrod Shilts. The funders had no role in study design, data collection and analysis, decision to publish, or preparation of the manuscript.

### Grant Disclosures

The following grant information was disclosed by the authors:
NIH: R35GM124685, 1U2COD023196, U01HG009086.
SyBBURE Searle Undergraduate Research Program.

## Competing Interests

The authors declare there are no competing interests.

## Author Contributions

- Jarrod Shilts and Jacob J. Hughey conceived and designed the experiments, performed the experiments, analyzed the data, wrote the paper, prepared figures and/or tables, reviewed drafts of the paper.
- Guanhua Chen conceived and designed the experiments, reviewed drafts of the paper.

## Data Availability

Figshare: https://doi.org/10.6084/m9.figshare.4906745.

An R package to calculate the CCD and ΔCCD is available at https://github.com/hugheylab/deltaccd. A web application is available at https://hugheylab.shinyapps.io/deltaccd.

## Supplemental Information

Supplemental information for this article can be found online at http://dx.doi.org/10.7717/peerj.4327#supplemental-information.

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
