# Peer review of "Evidence for widespread dysregulation of circadian clock progression in human cancer"

_PeerJ, doi:10.7717/peerj.4327_

## Round 0.1 · original submission · Major Revisions

The reviewers have brought up a fair number of questions regarding both the pipeline as well as quantification and associated conclusions. Please address these concerns.

Reviewer 1 ·

Basic reporting

no comment

Experimental design

no comment

Validity of the findings

See bellow

Additional comments

The authors used a comprehensive bioinformatics pipeline, including a newly created metric-delta clock correlation distance, to screen publically available data on healthy tissues for mouse, and to subsequently generate a signature of clock gene co-expression. Using their methodology, the authors concluded that clock dysregulation is a common treat in human cancer which is in agreement with other published studies.

The bioinformatics and statistical analysis presented in this study is solid and supports the conclusions of the paper. The established of a web tool will be useful and will contribute to facilitate the application of the methodology presented. The manuscript is of broad interest, the same sort of pipeline can be applied also to address other research questions, beyond the cancer vs healthy comparison.
I would therefore recommend publication in Peer J.

Nevertheless, a few points should be improved and/or better explained before publication, as listed bellow:
1 - A more detailed scheme of the pipeline used would facilitate the understanding of the methodology. In Figure 4A the authors have such a pipeline which could be completed by indicating the different levels/types of biological samples use. This would be particularly relevant since the authors use mouse samples to generate a signature for clock genes, this is later validated in human “healthy” samples, and in a next step used to analyse human cancer samples. This procedure should also be included in the scheme of Figure 4A.

2 - The authors include the cell line U2OS, a human osteosarcoma cell line, in the group of healthy human tissues – why? This sample should not have been included in this group. The authors must justify why they have decided to do so, and to which extent does this sample influence the overall results.

3 – The authors should better explain how their tool can be used to distinguish a phase shift from an expression change, between samples, for the core-clock genes.

4 – In line 348 the authors mention that tumour samples have a “weaker pattern”, this should be better explained since it is not clear what “weak” means, in this context.

5 - Figure 1A and Figure 2A show the expression of clock genes in human and mouse. The plots should be at least partially comparable, however Figure 1A uses a normalization in the range (1,-1), but Figure 2A uses a normalization in the range (2,-2; 3, -2). There is no apparent reason to do this and the same normalization should be used. Figure 2A is very difficult to interpret because the clustering is not well visible, it seems to be quite poor.

6 - Figure 3A should be revised, the samples have very different sizes and this makes interpretation difficult.

7 – The authors calculate a new score of delta-CCD, but have only one figure (Figure 4B) with this score. However, the figure is not very good at supporting the conclusion that the dysregulation of the circadian clock is a common mean by which human cancers achieve unrestrained growth and division. If the dysregulation would be so common for cancers, shouldn’t there be many high scores and a few low scores or at least a clear prevalence of high scores?

8 – The data presented does point to a dysregulation of the clock in tumours, but this bioinformatics analysis provides no evidence for a dysregulated clock as a means of achieving “unrestrained growth and division” (line 533). The authors should better justify this statement, eventually reformulate it.

Reviewer 2 ·

Basic reporting

Overall, the manuscript from Shilts and colleagues is well written and a well‐designed study with some major and minor cause for revision.
- Line 87: “In addition, although CYCLOPS can be used to infer rhythmicity in the expression of individual genes, it is not designed to quantify the differences in circadian clock function as a whole between multiple conditions.”. This sentence is unclear to me. What are differences in clock functions?
- Figure 1A shows Zeitgeber time, but in the description it is stated that samples were not labelled with the time of day (line 273). Are these predicted/estimated ZTs?
- Fig 1A is redundant with Fig. S3.
- Line 317: “The patterns of clock gene co-expression in human tissues and cells were similar to the patterns in mice (Fig. 2 and Fig. S4),..” The description of comparison of patterns should be made more clear. Therefore it would be helpful to state which Figures to compare with what, since in this case Figure 2 and S4 are not the ones to be compared to understand the descriptions of the results, rather Figure 1 and Fig. 2. This also applies to other comparisons.
- Line 322: “The strong pattern in human skin was due to clock gene co-expression both between the three time-points (9:30 am, 2:30 pm, and 7:30 pm) and between individuals at a given time-point (Fig. S6).” The Figure is labelled 14:30 and 19:30 instead of 2:30 or 7:30.
- Line 322: The description order of Fig. S6 needs to be the same as illustrated in the Figure.
- Description of Figure 2 are not in line with the test in the results. Fig 2 and Fig 2C have been described, but not Figure 2A, B.
- What is the difference between Fig 2 and S7? (ZTs?) Where is the description of lung and liver samples in Fig. 2?
- Line 391: “To better understand the nature of clock dysregulation in tumors, we compared clock gene expression in human cancer and in knockouts of clock genes in mice.”. Describe what tissue was used.
- Line 426: “In contrast, human tumor samples tended to show similar or somewhat higher MAD compared to non-tumor samples.”. Please comment on the fact that higher MAD was found in tumor tissues, which may indicate enhanced rather than reduced rhythmicity in tumor tissues compared to non-tumor tissues.
- Line 428: Taken together with the differential expression analysis, these results imply that clock dysregulation in human cancer is not due solely to loss of activity of one or more core clock genes. “. This sentence or the conclusion is unclear to me.
- Line 451: Remarkably, we observed a consistent pattern, in which genes with an acrophase near tended π to show the strongest reductions in expression in tumor samples (Fig. 6C, P ≤ 0.001 of non-zero amplitude by permutation testing for each dataset). Using Phase Set Enrichment Analysis (Zhang et al., 2016), we found that these genes were particularly enriched for involvement in G protein-coupled receptor signaling (Table S4).” This sentence is unclear to me. Which results support this?


Figure legends:
- abbreviations need to be xexplained even in figs
- S6: “Pattern of clock gene expression in human skin is due to both time-of-day variation and inter-individual variation.”. This sentence does not make sense to me.
- S14: The header is incorrect: Cry1/Cry2 KO are in the bottom panel not the top as indicated on the right. I suggest to write the specific genes in the header and the WT or KO on the right.

Experimental design

Experiments have been designed carefully and are mostly well described.
- Line 155: A brief description of ComBat would have been helpful to understand the procedure to reduce batch effects.
- Line 162: It would be important to know for which data sets the “strength of circadian rhythmicity” has been calculated.
- Line 232: A short explanation on what “gene modules” are would be helpful for the reader?

Validity of the findings

1. The strength of the described work lays in the quantification of similarities between gene expressions patterns of control and tumor tissues. However this technique has not been applied to all parts of the study. It puzzles me that the technique is only introduced at the last third of the manuscript, but not from the beginning on. While reading the manuscript my major criticism was the lack of quantification for the entire first half of the results, quantification was missing although the technique was available.
A statement such as “weak or absent” cannot be concluded from these results, since no quantification has been done. Similar descriptions of results have been made in the entire results section. Sentences in the results section such as “In non-tumor samples from most cancer types, we observed a similar pattern of clock gene co-expression as in the healthy mouse and human datasets (Fig. 3A-B and Fig. S8A) or “In contrast, in tumor samples from each cancer type, the pattern was weak or absent.” (line 348) need to be rewritten.
The introduction of the quantification for the very first dataset and then for all datasets following would strongly improve the manuscript and support/validate statements like “we observe the same trend of expression patter” or “we confirm our findings in additional human organs” such as in line 328 or
- line 317: “The patterns of clock gene co-expression in human tissues and cells were similar to the patterns in mice (Fig. 2 and Fig. S4),..”
- line 319-326: ” The pattern was less distinct in human blood (Fig. S5), likely because several clock genes show weak or no rhythmicity in expression in blood cells (Hughey & Butte, 2016). The strong pattern in human skin was due to clock gene co-expression both between the three time-points (9:30 am, 2:30 pm, and 7:30 pm) and between individuals at a given time-point (Fig. S6). Compared to co-expression patterns in mouse organs and human skin, those in human brain were somewhat weaker, which is consistent with the weaker circadian rhythmicity for clock genes in those two brain-specific datasets (Hughey & Butte, 2016).”
- Line 303: “In each dataset, the correlation pattern was preserved in both daytime and nighttime samples (Fig. S3).” This needs to be quantified before such a conclusion can be made.
-

2. Describing a similarity between species is impossible without quantification.
- Line 317: “The patterns of clock gene co-expression in human tissues and cells were similar to the patterns in mice (Fig. 2 and Fig. S4), consistent with our previous findings of similar relative phasing of clock gene expression in mice and humans (Hughey & Butte, 2016). The pattern was less distinct in human blood (Fig. S5), likely because several clock genes show weak or no rhythmicity in expression in blood cells (Hughey & Butte, 2016). The strong pattern in human skin was due to clock gene co-expression both between the three time-points (9:30 am, 2:30 pm, and 7:30 pm) and between individuals at a given time-point (Fig. S6). Compared to co- expression patterns in mouse organs and human skin, those in human brain were somewhat weaker, which is consistent with the weaker circadian rhythmicity for clock genes in those two brain-specific datasets (Hughey & Butte, 2016).”. There is not enough evidence presented that human samples are similar to mouse samples. Mouse patterns are rather strong, while human co-expression patterns are rather weak. This is not restricted to only blood samples as indicated by the authors, rather applies to all samples excluding from skin. The clock in humans and thereby gene expression amplitudes are not generally weaker. How do the author’s explain the differences in the expression profile strength between mice and humans? Saying they are similar as stated in line 317 is incorrect, since there are major differences in the contrast between both species. A quantification would give room for better phrasing, however to argue something is more similar to another can only be made, when a negative controls exist.
- Line 346: “In non-tumor samples from most cancer types, we observed a similar pattern of clock gene co-expression as in the healthy mouse and human datasets (Fig. 3A-B and Fig. S8A). In contrast, in tumor samples from each cancer type, the pattern was weaker or absent. We observed the same trend when we restricted our analysis to only matched samples, i.e., samples from 350 patients from whom both non-tumor and tumor samples were collected (Fig. S9).” A well defined quantification (including positive and negative control) is needed to state such a conclusion. For example (line 354) the described perturbation in Fig. 8B is not obvious to me. I would need more evidence including a quantification.


3. Another major point addresses the validation of conclusion made for the functionality of the circadian clock. As the author’s state in the introduction and in the discussion that the method introduced does not rely on time points, a conclusion about the functionality or rhythmicity of the clock is impossible to make. The results in the manuscript describe a specific clock gene pattern at a specific time point, not throughout the circadian cycle. Therefore sentences describing the rhythm or the circadian aspect of the clock have to be re-phrased. There are many sentences in the entire manuscript. Some examples are:
- the title of the manuscript “Widespread dysregulation of the circadian clock in human Cancer”. No statement can be made about the circadian clock status since no circadian/rhythm analysis have been performed. I rather suggest “altered co-expression profiles of clock genes”.
- Line 28: “Furthermore, by analyzing a large set of genes previously inferred to be rhythmic in healthy human lung, we observed systematic changes in patterns of circadian gene expression in lung tumors.” Instead of “circadian gene expression” I suggest rather “clock gene expression”.
- Line 272: ” Such a pattern could be used to infer the activity of the clock, even in datasets
- 273 in which samples are not labeled with time of day (Fig. 1A).”.What does even the “The activity of the clock” means? The status the expression level of specific clock genes?
- line 304: “These results indicate that our approach can detect an active circadian clock in groups of ….:). I would rather suggest something like “These results indicate that our approach can detect a co-expression pattern at a specific time points, similar to the one detectable in data from tissues with a functional circadian clock.”
- Line 337: “To investigate circadian clock function in human cancer, we applied our approach to…”. Circadian clock function cannot be investigated with these kind of analysis.
- needs to be changed accordingly
- line 378: ” Overall, these results suggest that the circadian clock is dysregulated in a range of human cancers.”. The presented data do not justify a conclusion like that, because the circadian aspect is irrelevant for the applied method.
- Line 417: “In the clock gene knockouts, rhythmic expression of most clock genes was reduced or lost (Fig. S16).” No rhythm analysis has been performed, so no conclusion about it can be drawn here. Same holds true for the legend of Fig. S16!
- Line 455: “Taken together, these findings suggest that dysregulation of the circadian clock in human lung cancer is accompanied by systematic changes in broader circadian gene expression.”
- Line 469: “Our approach for detecting a functional clock relies on three principles.”. The functionality of the circadian clock cannot be detected by the applied approach, rather a “normal” phase-relationship between clock genes, whereby “normal” means compared to a previously defined control pattern.
- Line 480: “Altogether, these principles enable our method to detect dysregulation of the clock in groups of samples whose times of acquisition are unknown and whose coverage of the 24-h cycle is incomplete.”.
- The conclusion needs to be re-phrased accordingly.

Additional comments

Apart from the changes I have suggested for the quantification, I enjoyed reading the manscuript and the idea behind the introduced analysis is well-thought-out and of importance for the cancer field.

---

## Round 0.2 · Minor Revisions

The consensus is that the manuscript has improved significantly, though both reviewers have a few minor concerns/revisions. Please consider these as part of a minor revision.

Reviewer 1 ·

Basic reporting

The article meets the necessary standards.

Experimental design

See general comments.

Validity of the findings

See general comments.

Additional comments

In this revised form of the manuscript the authors have addressed most of my comments, as well as the comments from the other reviewer. I find this study very well-thought, of overall interest, and only have a few remaining minor comments, which can be easily addressed before publication of the manuscript.

My main worry goes back to my previous comment regarding the usage of U2OS (human osteosarcoma) as “normal clock” and the subsequent statement by the authors in the rebuttal letter “we believe stronger rhythms in tumour samples are unlikely”.
The authors consider that the clock in U2OS cells is “strong” (I would agree). Furthermore, other studies (Relogio et al Plos Genetics 2014, cited by the authors in this manuscript) also show the existence of different clock phenotypes (including strong clocks) within the same cancer type. Such strong rhythms observed in cancer cells contradict the statement “we believe stronger rhythms in tumour samples are unlikely”.
Altogether, it is not trivial to associate the “quality of the clock” with cancer, even though compelling evidence points to a strong clock-cancer correlation with which I would also tend to agree. As such, I would suggest that the authors are more critical regarding the interpretation of the data, and rewrite a few sentences in the discussion.
1-For e.g. in the first paragraph of the discussion the authors write “…we discovered that clock progression may be perturbed in multiple types of human cancer” what do the authors mean by clock progression?
2-On pg 13 of the discussion the authors write “…the dysregulation of clock progression in cancer maybe readily reversible”. Again, this is a rather strong statement based on the fact that some cancer cells lines have a strong clock, maybe these cells have always kept a “healthy clock”.
3-Finally, to help reinforcing the message of the paper regarding the dysregulation of the circadian clock in cancer, the authors could add the following (very recent) references to the discussion, which strengthens the conclusions and perspectives presented:
The circadian clock as a tumour suppressor:
a) Ticking time bombs: connections between circadian clocks and cancer. Lamia KA. F1000Res. 2017
b) The Ink4a/Arf locus operates as a regulator of the circadian clock modulating RAS activity. El-Athman R, et al. PLoS Biol. 2017
c) Circadian disruption promotes tumor growth by anabolic host metabolism; experimental evidence in a rat model. Guerrero-Vargas et al BMC Cancer. 2017
The circadian clock in treatment:
d) The tumor circadian clock: a new target for cancer therapy? Kiessling S, Cermakian N. Future Oncol. 2017
e) Dosing time dependent in vitro pharmacodynamics of Everolimus despite a defective circadian clock. Zhang Y, et al Cell Cycle. 2017

Reviewer 2 ·

Basic reporting

Overall, the manuscript from Shilts and colleagues has improved significantly since the last submission. It is well written and a well‐designed study with some minor cause for revision.

Results
- In the results section the Supplemental Figure citations (from Fig S6, S7, S8) are not in order. Fig. S6A, B are described first, then Fig. S7 and S8 and finally Fig. S6C, D. Either Fig. S6C, D becomes the new Figure S9, or its decription needs to be reorganized.


Figure legends:
- abbreviations need to be explained even in figs, such as ZT = Zeitgeber time
- statistical values such as p<0.005 for the p-values such as in Fig. 2 are not standard. p<0.05, P<0.01, P<0.01 and P<0.001 is the standard.

Experimental design

no comment

Validity of the findings

In the new version of the manuscript the authors have included my suggestion and added analysis to the relevant sections and rewrote the conclusion about the functionality of the clock to my satisfaction. Although I have some minor comments:
- Line 365- ongoing: I am puzzled by the fact that the human tumor and non-tumor data have always been compared with the reference data set obtained on mouse tissue, although human datasets have been available and even more with indication about the Zeitgeber time. As indicated in the results Fig. 1, human data show rather similar pattern as results obtained from mice, but as indicated elsewhere co-expression pattern in 2 out of 6 human data sets are not significantly similar to the ones obtained from mice. Shouldn’t this facts have lead the authors to generate a new reference set using the available human data set with time information to be used for all the following comparisons between human g non-tumor and tumor data?
- Using tissues from clock knockout mice with only loss of one clock gene (excluding Bmal1), such as Clock or Reverba should not be pooled with data sets obtained from mice with either deletion of two essential clock genes or Bmal1, which cause arrhythmicity. Complete circadian disruption is only accomplished by genetic disruption of Bmal1 or two essential clock components such as Cry1/2-/-, whereas RevErba-/- mice and Clock-/- mice are still rhythmic, but with a different tau! Thus, in their own circadian day length the phase relationship of clock genes to each other collected from samples during the 24 day is likely changed. This may explain the differences in Fig. S13 and S14 within the knockout cohort and also the results described in line 401-406 regarding fig. 3A rather than the small sample size as indicated by the authors. I would rather recommend under this circumstances to keep it simple and stick to completely arrhythmic mice or organs.
- Line 408-416: The relevance to compare human cancer and clock knockout data sets is not intriguing, since human cancer tissues usually show disrupted clock gene expression, but usually not a complete loss of any of the core clock components. The difference between clock knockout and suppression or upregulation in tumor tissues which results in arrhythmicity of the expression, is a completely different situation and not comparable. Literature reports strong up or down-regulation of essential clock genes, which then influences the status of interconnected clock genes, however, I am not aware of reports indicating complete loss of clock genes due to mutation in human tumor cells. Therefore this comparison is invalid or irrelevant and very artificial to me and the conclusion e.g. indicated in line 106: ” Our findings suggest that dysregulation of circadian clock progression is present in a wide range of human cancers, is not caused solely by the loss of core clock genes, and is accompanied by systematic changes in broader circadian gene expression.“ seems a wrong interpretation and misleading.

---

## Round 0.3 · accepted · Accept

Thank you for fully addressing the reviewers' questions and comments.